# Any-to-Any Generation via Composable Diffusion

**Zineng Tang**[1]*    **Ziyi Yang**[2]†    **Chenguang Zhu**[2]‡    **Michael Zeng**[2]    **Mohit Bansal**[1]†

[1]University of North Carolina at Chapel Hill
[2]Microsoft Azure Cognitive Services Research
https://codi-gen.github.io

## Abstract

We present Composable Diffusion (CoDi), a novel generative model capable of generating any combination of output modalities, such as language, image, video, or audio, from any combination of input modalities. Unlike existing generative AI systems, CoDi can generate multiple modalities in parallel and its input is not limited to a subset of modalities like text or image. Despite the absence of training datasets for many combinations of modalities, we propose to align modalities in both the input and output space. This allows CoDi to freely condition on any input combination and generate any group of modalities, even if they are not present in the training data. CoDi employs a novel composable generation strategy which involves building a shared multimodal space by bridging alignment in the diffusion process, enabling the synchronized generation of intertwined modalities, such as temporally aligned video and audio. Highly customizable and flexible, CoDi achieves strong joint-modality generation quality, and outperforms or is on par with the unimodal state-of-the-art for single-modality synthesis. The project page with demonstrations and code is at https://codi-gen.github.io/

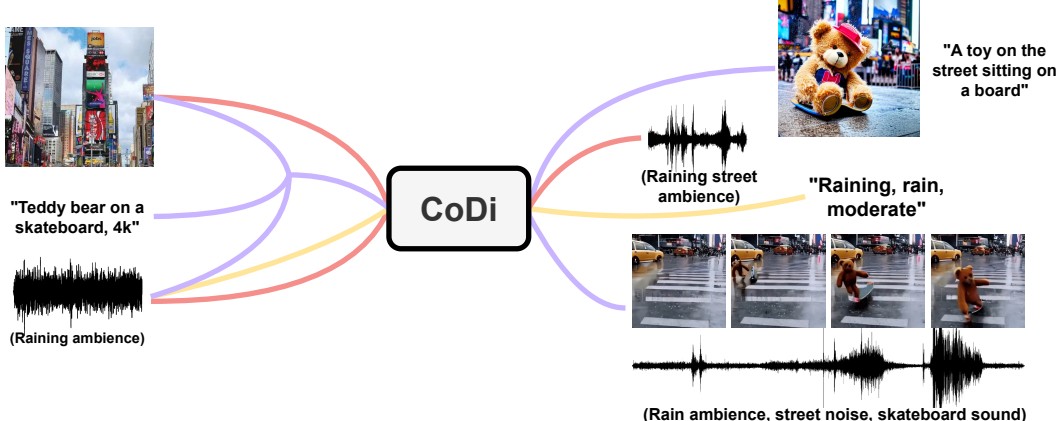

Figure 1: CoDi can generate various (joint) combinations of output modalities from diverse (joint) sets of inputs: video, image, audio, and text (example combinations depicted by the colored arrows).

---

*Work done at Microsoft internship and UNC.

†Corresponding authors: ziyiyang@microsoft.com, mbansal@cs.unc.edu

‡Work done while at Microsoft.

37th Conference on Neural Information Processing Systems (NeurIPS 2023).

# 1 Introduction

Recent years have seen the rise of powerful cross-modal models that can generate one modality from another, e.g. text-to-text [6, 37], text-to-image [13, 19, 22, 41, 44], or text-to-audio [23, 33]. However, these models are restricted in their real-world applicability where multiple modalities coexist and interact. While one can chain together modality-specific generative models in a multi-step generation setting, the generation power of each step remains inherently limited, and a serial, multi-step process can be cumbersome and slow. Moreover, independently generated unimodal streams will not be consistent and aligned when stitched together in a post-processing way (e.g., synchronized video and audio). The development of a comprehensive and versatile model that can generate any combination of modalities from any set of input conditions has been eagerly anticipated, as it would more accurately capture the multimodal nature of the world and human comprehension, seamlessly consolidate information from a wide range of sources, and enable strong immersion in human-AI interactions (for example, by generating coherent video, audio, and text description at the same time).

In pursuit of this goal, we propose Composable Diffusion, or CoDi, the first model capable of simultaneously processing and generating arbitrary combinations of modalities as shown in Fig. 1. Training a model to take any mixture of input modalities and flexibly generate any mixture of outputs presents significant computational and data requirements, as the number of combinations for the input and output modalities scales exponentially. Also aligned training data for many groups of modalities is scarce or even non-existent, making it infeasible to train with all possible input-output combinations. To address this challenge, we propose to align multiple modalities in both the input conditioning (Section 3.2) and generation diffusion step (Section 3.4). Furthermore, a proposed "Bridging Alignment" strategy for contrastive learning (Section 3.2) allows us to efficiently model the exponential number of input-output combinations with a linear number of training objectives.

Building a model with any-to-any generation capacity with exceptional generation quality requires comprehensive model design and training on diverse data resources. Therefore, we build CoDi in an integrative way. First, we train a latent diffusion model (LDM) for each modality, e.g., text, image, video, and audio. These models can be trained in parallel independently, ensuring exceptional single-modality generation quality using widely available modality-specific training data (i.e., data with one or more modalities as input and one modality as output). For conditional cross-modality generation, such as generating images using audio+language prompts, the input modalities are projected into a shared feature space (Section 3.2), and the output LDM attends to the combination of input features. This multimodal conditioning mechanism prepares the diffusion model to condition on any modality or combination of modalities without directly training for such settings.

The second stage of training enables the model to handle many-to-many generation strategies that involve simultaneously generating arbitrary combinations of output modalities. To the best of our knowledge, CoDi is the first AI model with this capability. This is achieved by adding a cross-attention module to each diffuser, and an environment encoder $V$ to project the latent variable of different LDMs into a shared latent space (Section 3.4). Next, we freeze the parameters of the LDM, training only the cross-attention parameters and $V$. Since the environment encoder of different modalities are aligned, an LDM can cross-attend with any group of co-generated modalities by interpolating the representation's output by $V$. This enables CoDi to seamlessly generate any group of modalities, without training on all possible generation combinations. This reduces the number of training objectives from exponential to linear.

We demonstrate the any-to-any generation capability of CoDi, including single-to-single modality generation, multi-condition generation, and the novel capacity of joint generation of multiple modalities. For example, generating synchronized video and audio given the text input prompt; or generating video given a prompt image and audio. We also provide a quantitative evaluation of CoDi using eight multimodal datasets. As the latest work from Project i-Code [55] towards Composable AI, CoDi exhibits exceptional generation quality across assorted scenarios, with synthesis quality on par or even better than single to single modality SOTA, e.g., audio generation and audio captioning.

# 2 Related Works

**Diffusion models (DMs)** learn the data distribution by denoising and recovering the original data. Deep Diffusion Process (DDP) [45] adopts a sequence of reversible diffusion steps to model image

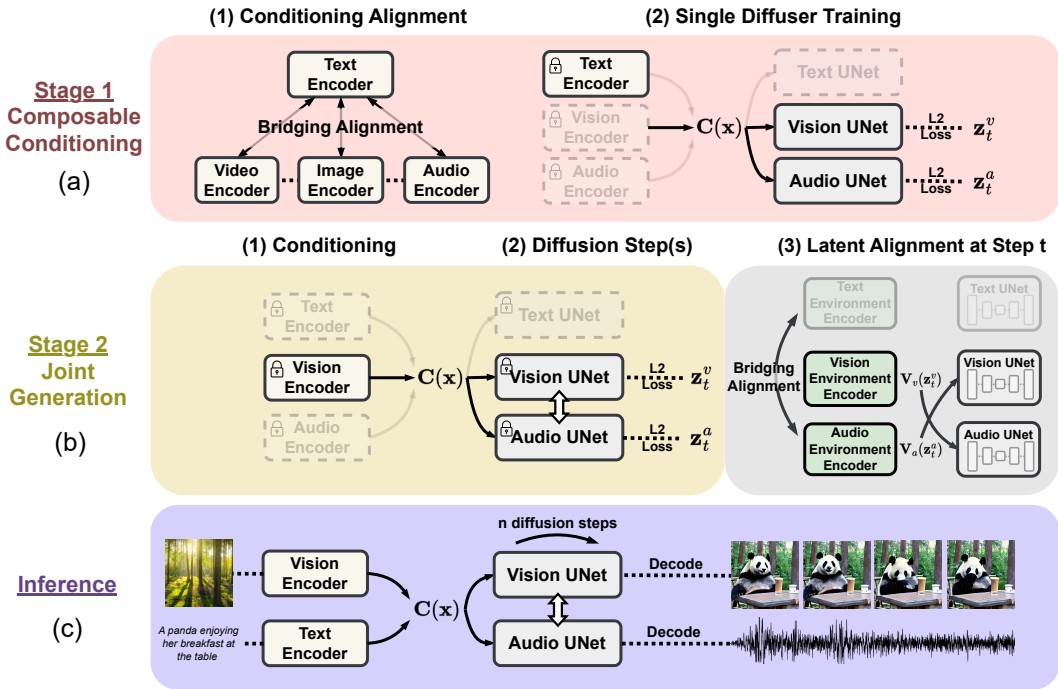

Figure 2: CoDi model architecture: (a) We first train individual diffusion model with aligned prompt encoder by "Bridging Alignment"; (b) Diffusion models learn to attend with each other via "Latent Alignment"; (c) CoDi achieves any-to-any generation with a linear number of training objectives.

probability distribution. It uses a reversible encoder to map the input image to a latent space and a decoder to map the latent variables to an output image. Denoising diffusion probabilistic model (DDPM) [20] uses a cascade of diffusion processes to gradually increase the complexity of the probability density function model. At each step, the model adds noise to the input image and estimates the corresponding noise level using an autoregressive model. This allows the model to capture the dependencies between adjacent pixels and generate high-quality images. Score-based generative models (SOG) [46] use the score function to model the diffusion process. [40] generates high-fidelity images conditioned on CLIP representations of text prompts. Latent diffusion model (LDM) [41] uses a VAE to encode inputs into latent space to reduce modeling dimension and improves efficiency. The motivation is that image compression can be separated into semantic space by a diffusion model and perceptual space by an autoencoder. By incorporating temporal modeling modules and cascading model architectures, video diffusion models have been built upon image diffusers to generate temporally consistent and inherent frames[14, 19, 21, 44]. Diffusion models have also been applied to other domains, such as generating audio from text and vision prompts[23, 33].

**Multimodal modeling** has experienced rapid advancement recently, with researchers striving to build uniform representations of multiple modalities using a single model to achieve more comprehensive cross-modal understanding. Vision transformers [11], featuring diverse model architectures and training techniques, have been applied to various downstream tasks such as vision Q&A and image captioning. Multimodal encoders have also proven successful in vision-language [1, 8, 57], video-audio [47] and video-speech-language [55, 56] domains. Aligning data from different modalities is an active research area [12, 38], with promising applications in cross-modality retrieval and building uniform multimodal representations [33, 35, 41].

## 3 Methodology

### 3.1 Preliminary: Latent Diffusion Model

Diffusion models (DM) represent a class of generative models that learn data distributions $p(\boldsymbol{x})$ by simulating the diffusion of information over time. During training, random noise is iteratively added

to $\boldsymbol{x}$, while the model learns to denoise the examples. For inference, the model denoises data points sampled from simple distributions such as Gaussian. Latent diffusion models (LDM) [41] learn the distribution of the latent variable $\boldsymbol{z}$ corresponding to $\boldsymbol{x}$, significantly reducing computational cost by decreasing the data dimension.

In LDM, an autoencoder is first trained to reconstruct $\boldsymbol{x}$, i.e., $\hat{\boldsymbol{x}} = D(E(\boldsymbol{x}))$, where $E$ and $D$ denote the encoder and decoder, respectively. The latent variable $\boldsymbol{z} = E(\boldsymbol{x})$ is iteratively diffused over time steps $t$ based on a variance schedule $\beta_1, \ldots, \beta_T$, i.e., $q(\boldsymbol{z}_t|\boldsymbol{z}_{t-1}) = \mathcal{N}(\boldsymbol{z}_t; \sqrt{1 - \beta_t}\boldsymbol{z}_{t-1}, \beta_t \boldsymbol{I})$ [20, 45].

The forward process allows the random sampling of $\boldsymbol{z}_t$ at any timestep in a closed form [20, 45]: $\boldsymbol{z}_t = \alpha_t \boldsymbol{z} + \sigma_t \boldsymbol{\epsilon}$, where $\boldsymbol{\epsilon} \sim \mathcal{N}(0, I)$, $\alpha_t := 1 - \beta_t$ and $\sigma_t := 1 - \prod_{s=1}^{t} \alpha_s$. The diffuser learns how to denoise from $\{\boldsymbol{z}_t\}$ to recover $\boldsymbol{z}$. Following the reparameterization method proposed in [20], the denoising training objective can be expressed as [41]:

$$\mathcal{L}_D = \mathbb{E}_{\boldsymbol{z},\boldsymbol{\epsilon},t} \|\boldsymbol{\epsilon} - \boldsymbol{\epsilon}_\theta(\boldsymbol{z}_t, t, C(\boldsymbol{y}))\|_2^2. \tag{1}$$

In data generation, the denoising process can be realized through reparameterized Gaussian sampling:

$$p(\boldsymbol{z}_{t-1}|\boldsymbol{z}_t) = \mathcal{N}\left(\boldsymbol{z}_{t-1}; \frac{1}{\sqrt{\alpha_t}}\left(\boldsymbol{z}_t - \frac{\beta_t}{\sqrt{\sigma_t}}\boldsymbol{\epsilon}_\theta\right), \beta_t \boldsymbol{I}\right). \tag{2}$$

In $\mathcal{L}_D$, the diffusion time step $t \sim \mathcal{U}[1, T]$; $\boldsymbol{\epsilon}_\theta$ is a denoising model with UNet backbone parameterized by $\theta$; $\boldsymbol{y}$ represents the conditional variable that can be used to control generation; $C$ is the prompt encoder. The conditioning mechanism is implemented by first featurizing $\boldsymbol{y}$ into $C(\boldsymbol{y})$, then the UNet $\boldsymbol{\epsilon}_\theta$ conditions on $C(\boldsymbol{y})$ via cross-attention, as described in [41]. Distinct from previous works, our model can condition on any combinations of modalities of text, image, video and audio. Details are presented in the following section.

## 3.2 Composable Multimodal Conditioning

To enable our model to condition on any combination of input/prompt modalities, we align the prompt encoder of text, image, video and audio (denoted by $C_t$, $C_i$, $C_v$, and $C_a$, respectively) to project the input from any modality into the same space. Multimodal conditioning can then be conveniently achieved by interpolating the representations of each modality $m$: $C(x_t, x_i, x_v, x_a) = \sum_m \alpha_m C(m)$ for $m \in x_t, x_i, x_v, x_a$, with $\sum_m \alpha_m = 1$. Through simple weighted interpolation of aligned embeddings, we enable models trained with single-conditioning (i.e., with only one input) to perform zero-shot multi-conditioning (i.e., with multiple inputs). This process is illustrated in Fig. 2 (a)(2).

Optimizing all four prompt encoders simultaneously in a combinatorial manner is computationally heavy, with $\mathcal{O}(n^2)$ pairs. Additionally, for certain dual modalities, well-aligned paired datasets are limited or unavailable e.g., image-audio pairs. To address this challenge, we propose a simple and effective technique called "Bridging Alignment" to efficiently align conditional encoders. As shown in Fig. 2 (a)(1), we choose the text modality as the "bridging" modality due to its ubiquitous presence in paired data, such as text-image, text-video, and text-audio pairs. We begin with a pretrained text-image paired encoder, i.e., CLIP [38]. We then train audio and video prompt encoders on audio-text and video-text paired datasets using contrastive learning, with text and image encoder weights frozen.

In this way, all four modalities are aligned in the feature space. As shown in Section 5.2, CoDi can effectively leverage and combine the complementary information present in any combination of modalities to generate more accurate and comprehensive outputs. The high generation quality remains unaffected with respect to the number of prompt modalities. As we will discuss in subsequent sections, we continue to apply Bridging Alignment to align the latent space of LDMs with different modalities to achieve joint multimodal generation.

## 3.3 Composable Diffusion

Training an end-to-end anything-to-anything model requires extensive learning on various data resources. The model also needs to maintain generation quality for all synthesis flows. To address these challenges, CoDi is designed to be composable and integrative, allowing individual modality-specific models to be built independently and then smoothly integrated later. Specifically, we start by

independently training image, video, audio, and text LDMs. These diffusion models then efficiently learn to attend across modalities for joint multimodal generation (Section 3.4) by a novel mechanism named "latent alignment".

**Image Diffusion Model.** The image LDM follows the same structure as Stable Diffusion 1.5 [41] and is initialized with the same weights. Reusing the weights transfers the knowledge and exceptional generation fidelity of Stable Diffusion trained on large-scale high-quality image datasets to CoDi.

**Video Diffusion Model.** To model the temporal properties of videos and simultaneously maintain vision generation quality, we construct the video diffuser by extending the image diffuser with temporal modules. Specifically, we insert pseudo-temporal attention before the residual block [13]. However, we argue that pseudo-temporal attention only enables video frames to globally attend to each other by flattening the pixels (height, width dimension) to batch dimension, resulting in a lack of cross-frame interaction between local pixels. We argue that this results in the common temporal-inconsistency issue in video generation that locations, shapes, colors, etc. of objects can be inconsistent across generated frames. To address this problem, we propose adapting the latent shift method [2] that performs temporal-spatial shifts on latent features in accordance with temporal attention. We divide the video by the hidden dimension into $k = 8$ chunks, and for each chunk $i = 0$ to 7, we shift the temporal dimension forward by $i$ positions. Further details will be provided in the appendix.

**Audio Diffusion Model.** To enable flexible cross-modality attention in joint generation, the audio diffuser is designed to have a similar architecture to vision diffusers, where the mel-spectrogram can be naturally viewed as an image with 1 channel. We use a VAE encoder to encode the mel-spectrogram of audio to a compressed latent space. In audio synthesis, a VAE decoder maps the latent variable to the mel-spectrogram, and a vocoder generates the audio sample from the mel-spectrogram. We employ the audio VAE from [33] and the vocoder from [27].

**Text Diffusion Model.** The VAE of the text LDM is OPTIMUS [29], and its encoder and decoder are [9] and GPT-2 [39], respectively. For the denoising UNet, unlike the one in image diffusion, the 2D convolution in residual blocks is replaced with 1D convolution [53].

### 3.4 Joint Multimodal Generation by Latent Alignment

The final step is to enable cross-attention between diffusion flows in joint generation, i.e., generating two or more modalities simultaneously. This is achieved by adding cross-modal attention sublayers to the UNet $\epsilon_\theta$ (Fig. 2 (b)(2)). Specifically, consider a diffusion model of modality $A$ that cross-attends with another modality $B$. Let the latent variables of modalities $m_A$ and $m_B$ at diffusion step $t$ be denoted as $z_t^A$ and $z_t^B$, respectively. The proposed "Latent Alignment" technique is such that a modality-specific environment encoder $V_B$ first projects $z_t^B$ into a shared latent space for different modalities. Then, in each layer of the UNet for modality $A$, a cross-attention sublayer attends to $V_B(z_t^B)$. For the diffusion model of modality $A$, the training objective in Eq. (1) now becomes:

$$\mathcal{L}_{Cross}^A = \mathbb{E}_{\boldsymbol{z},\boldsymbol{\epsilon},t}\|\boldsymbol{\epsilon} - \boldsymbol{\epsilon}_{\theta_c}(\boldsymbol{z}_t^A, V_B(\boldsymbol{z}_t^B), t, C(\boldsymbol{y}))\|_2^2, \tag{3}$$

where $\theta_c$ denotes the weights of cross-attention modules in the UNet.

The training objective of $A + B$ joint generation is $\mathcal{L}_{Cross}^A + \mathcal{L}_{Cross}^B$. $V(\cdot)$ of different modalities are trained to be aligned with contrastive learning. Since $z_t^A$ and $z_t^B$ at any time step can be sampled with closed form in the diffusion process Section 3.1, one can conveniently train the contrastive learning together with $\mathcal{L}_{Cross}$. The purpose of $V$ is to achieve the generation of any combination of modalities (in polynomial) by training on a linear number of joint-generation tasks. For example, if we have trained the joint generation of modalities $A$, $B$, and $B$, $C$ independently, then we have $V_A(z_t^A)$, $V_B(z_t^B)$, and $V_C(z_t^C)$ aligned. Therefore, CoDi can seamlessly achieve joint generation of modalities $A$ and $C$ without any additional training. Moreover, such design automatically effortlessly enables joint generation of modalities $A$, $B$, and $C$ concurrently. Specifically, UNet of $A$ can cross-attend with the interpolation of $V_B(z_t^B)$, and $V_C(z_t^C)$, although CoDi has not been trained with such task.

As shown in Fig. 2(b)(3), we follow similar designs to the "Bridging Alignment" in training joint generation: (1) We first train the cross-attention weights in the image and text diffusers, as well

Table 1: Training tasks (CT stands for "contrastive learning" to align prompt encoders) and datasets with corresponding statistics. * denotes the number of accessible examples in the original datasets.

| Categories | Tasks | Datasets | # of samples | Domain |
|---|---|---|---|---|
| Image + Text | Image→Text, Text→Image Text→Image+Text | Laion400M [42] | 400M | Open |
| Audio + Text | Text→Audio, Audio→Text, Text→Audio+Text, Audio-Text CT | AudioSet [16] AudioCaps [24] Freesound 500K BBC Sound Effect | 900K* 46K 2.5M 30K | YouTube YouTube Public audio samples Authentic natural sound |
| Audiovisual | Image→Audio, Image→Video+Audio | AudioSet SoundNet [3] | 900K* 1.0M* | YouTube Flickr, natural sound |
| Video | Text→Video, Image→Video, Video-Text CT | Webvid10M [4] HD-Villa-100M [54] | 10.7M 100M | Short videos YouTube |

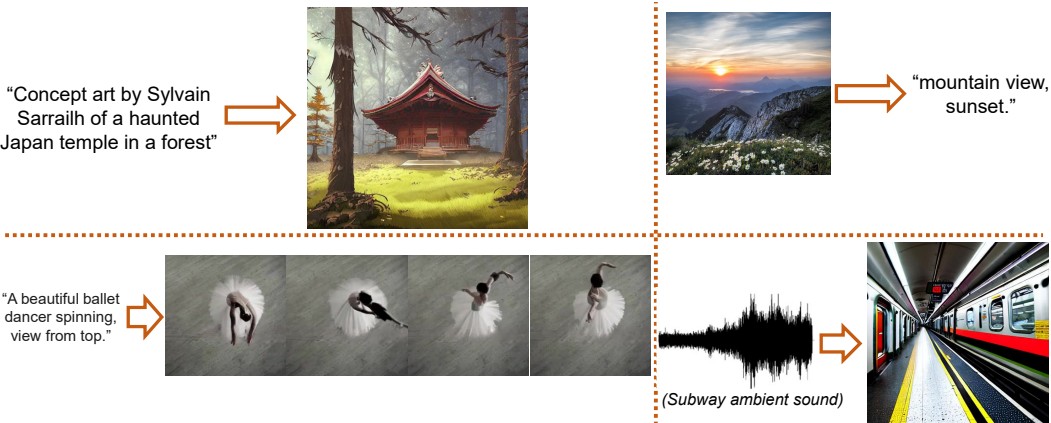

Figure 3: Single-to-single modality generation. Clockwise from top left: text→image, image→text, image→video, audio→image.

as their environment encoders $V$, on text-image paired data. (2) We freeze the weights of the text diffuser and train the environment encoder and cross-attention weights of the audio diffuser on text-audio paired data. (3) Finally we freeze the audio diffuser and its environment encoder, and train the joint generation of the video modality on audio-video paired data. As demonstrated in Section 5.3, although only trained on three paired joint generation tasks (i.e, Text+Audio, Text+Image, and Video+Audio), CoDi is capable of generating assorted combinations of modalities simultaneously that are unseen in training, e.g., joint image-text-audio generation in Fig. 5.

## 4 Experiments

### 4.1 Training Objectives and Datasets

We list training tasks of CoDi in Table 1, including single modality synthesis, joint multimodal generation, and contrastive learning to align prompt encoders. Table 1 provides an overview of the datasets, tasks, number of samples, and domain. Datasets are from the following domains: **image + text** (e.g. image with caption), **audio + text** (e.g. audio with description), **audio + video** (e.g. video with sound), and **video + text** (e.g. video with description). As one may have noticed, the language modality appears in most datasets and domains. This echos the idea of using text as the bridge modality to be able to extrapolate and generate new unseen combinations such as audio and image bridged by text, as mentioned in Section 3.2 and Section 3.4. Due to space limit, more details on training datasets and can be found in Appendix C, model architecture details in Appendix Appendix A.1, and training details in Appendix B.

**Image + Text.** We use a recently developed large-scale image caption dataset, Laion400M [42]. This image-text paired data allows us to train with tasks **text→image**, **image→text**, and the joint

Table 2: COCO-caption [32] FID scores for text-to-image generation.

| Method | FID ↓ |
|--------|-------|
| CogView [10] | 27.10 |
| GLIDE [36] | 12.24 |
| Make-a-Scene [15] | 11.84 |
| LDM [41] | 12.63 |
| Stable Diffusion-1.4 | 11.21 |
| Stable Diffusion-1.5 | 11.12 |
| Versatile Diffusion [53] | 11.10 |
| **CoDi (Ours)** | 11.26 |

Table 3: MSR-VTT text-to-video generation performance.

| Method | Zero-Shot | CLIPSIM ↑ |
|--------|-----------|-----------|
| GODIVA [50] | No | 0.2402 |
| NÜWA [51] | No | 0.2439 |
| CogVideo [22] | Yes | 0.2631 |
| Make-A-Video [44] | Yes | 0.3049 |
| Video LDM [5] | Yes | 0.2929 |
| **CoDi (Ours)** | Yes | 0.2890 |

Table 4: UCF-101 text-to-video generation performance.

| Method | IS (↑) | FVD (↑) |
|--------|--------|---------|
| CogVideo (Chinese) | 23.55 | 751.34 |
| CogVideo (English) | 25.27 | 701.59 |
| Make-A-Video | 33.00 | 367.23 |
| Video LDM | 33.45 | 550.61 |
| **CoDi (Ours)** | 32.88 | 596.34 |

Table 5: The comparison between our audio diffuser and baseline TTA generation models. Evaluation is conducted on AudioCaps test set. AS, AC, FSD, BBC, and SDN stand for AudioSet, AudioCaps, Freesound, BBC Sound Effect, and Soundnet.

| Model | Datasets | FD ↓ | IS ↑ | KL ↓ | FAD ↓ | OVL ↑ | REL ↑ |
|-------|----------|------|------|------|-------|-------|-------|
| Ground truth | - | - | - | - | - | 83.61 | 80.11 |
| DiffSound | AS + AC | 47.68 | 4.01 | 2.52 | 7.75 | 45.00 | 43.83 |
| AudioGen | AS + AC + 8 others | - | - | 2.09 | 3.13 | - | - |
| AudioLDM-L-Full | AS + AC + FSD + BBC | 23.31 | 8.13 | 1.59 | 1.96 | 65.91 | 65.97 |
| **CoDi (Ours)** | AS + AC + FSD + BBC + SDN | **22.90** | **8.77** | **1.40** | **1.80** | **66.87** | **67.60** |

Table 6: COCO image captioning scores comparison.

| Model | B@4 | METEOR | CIDEr |
|-------|-----|--------|-------|
| *Autoregressive Model* | | | |
| Oscar [31] | 36.58 | 30.4 | 124.12 |
| ClipCap [35] | 32.15 | 27.1 | 108.35 |
| OFA [49] | 44.9 | 32.5 | 154.9 |
| BLIP2 [30] | 43.7 | - | 145.8 |
| *Diffusion Model* | | | |
| DDCap [59] | 35.0 | 28.2 | 117.8 |
| SCD-Net [34] | 39.4 | 29.2 | 131.6 |
| **CoDi (Ours)** | 40.2 | 31.0 | 149.9 |

Table 7: AudioCaps audio captioning scores comparison.

| Model | SPIDEr | CIDEr | SPICE |
|-------|--------|-------|-------|
| AudioCaps [24] | 0.369 | 0.593 | 0.144 |
| BART-Finetune [17] | 0.465 | 0.753 | 0.176 |
| VALOR [7] | - | 0.741 | - |
| AL-MixGen [25] | 0.466 | 0.755 | 0.177 |
| **CoDi (Ours)** | **0.480** | **0.789** | **0.182** |

Table 8: MSRVTT video captioning scores comparison.

| Model | B@4 | METEOR | CIDEr |
|-------|-----|--------|-------|
| ORG-TRL [58] | 43.6 | 28.8 | 50.9 |
| MV-GPT [43] | 48.9 | 38.7 | 60.0 |
| GIT [48] | 54.8 | 33.1 | 75.9 |
| mPLUG-2 [52] | 57.8 | 34.9 | 80.3 |
| **CoDi (Ours)** | 52.1 | 32.5 | 74.4 |

generation of image and text. For the joint generation task, we propose to train with **text→image+text**, where the prompt text is the truncated image caption, and the output text is the original caption. Since the condition information is incomplete, the text and image diffuser will need to learn to attend with each other through the joint generation process.

**Audio + Text.** We curated a new dataset, Freesound 500K, by crawling 500K audio samples together with tags and descriptions from the Freesound website. We also use AudioSet [42] with 2 million human-labeled 10-second sound clips from YouTube videos and AudioCaps [24] with 46K audio-text pairs derived from the AudioSet dataset. Audio samples are clipped into 10-second segments for training purposes. The paired audio + text data enables us to train **text→audio**, **audio→text**, **text→audio + text** generation, and audio-text contrastive learning. Similar to image + text joint generation, in text→audio + text, text prompt is the truncated text, and the output is the original text.

**Video.** We use the following diverse and high-quality video datasets to train video generation and video prompt encoder. WebVid [4], a large-scale dataset of web videos together with descriptions; HD-Villa-100M [54] with high resolution YouTube videos of at least 720P. We perform **text→video** and video-text contrastive learning task with WebVid. We use HD-Villa-100M for **image→video** generation where the middle frame is the input image.

**Audiovisual.** Web videos are a natural aligned audio-video data resource. However, many existing datasets, e.g., ACAV100M [28], feature heavily on videos of human speech rather than natural sounds. Therefore, we leverage sound-oriented datasets AudioSet and SoundNet [3] for joint audio-video generation. For **image→audio + video**, we use the middle frame of the target video as the input prompt image. We also use the middle frame as the prompt input to train the model to generate the audio, i.e., **image→audio**.

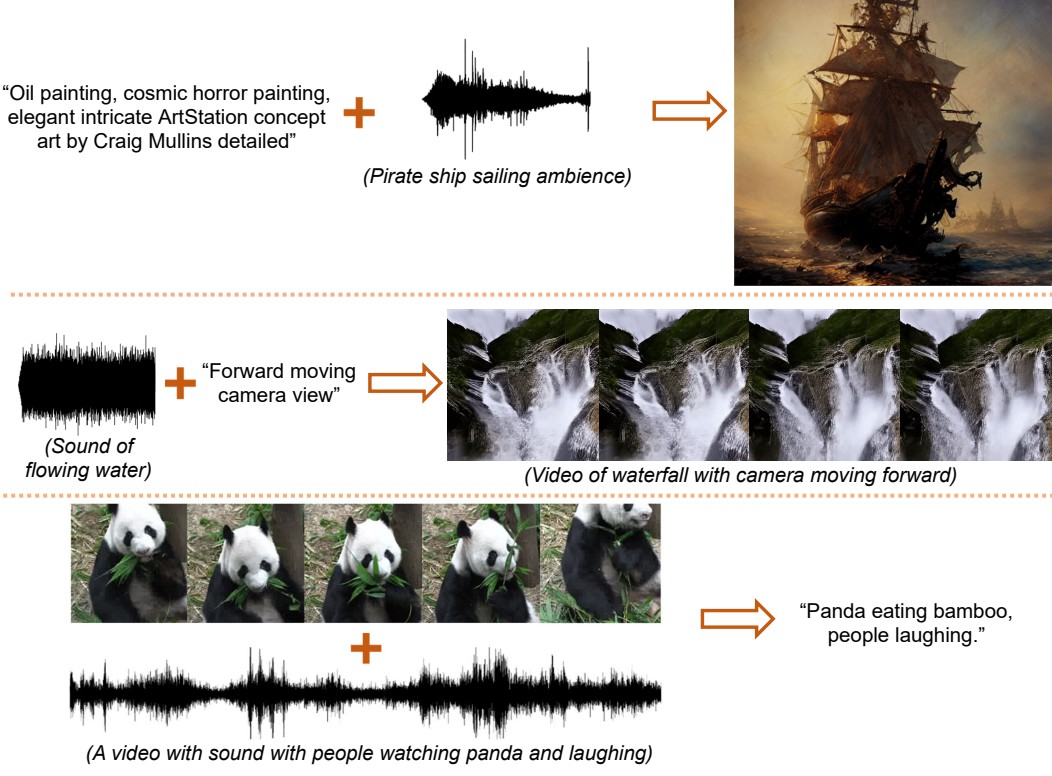

Figure 4: Generation with multiple input modality conditions. Top to bottom: text+audio→image, text+audio→video, video+audio→text.

## 5 Evaluation Results

In this section, we will evaluate the model generation quality in different settings including single modality generation, multi-condition generation, and multi-output joint generation. We provide both quantitative benchmarking on evaluation datasets as well as qualitative visualization demonstrations.

### 5.1 Single Modality Generation Results

We first show example demo in Fig. 3, where we present various single to single modality generation. Then, we evaluate the synthesis quality of the unimodal generation on text, image, video, and audio. CoDi achieves SOTA on audio captions and audio generation, as shown in Table 7 and Table 5. Notably for the first time in the field, CoDi, a diffusion-base model, exhibits comparable performance on image captioning with autoregressive transformer-based SOTA (Table 6). CoDi is the first diffusion-model based for video captioning Table 8. On image and video generation, CoDi performs competitively with state-of-the-art (Tables 2 to 4). This gives us strong starting points for multi-condition and multi-output generation that will be presented next in Section 5.2 and Section 5.3.

We demonstrate in Section 3.2 that CoDi is capable of integrating representation from different modalities in the generation. Thus, we first show multi-condition generation demo as shown in Fig. 4.

### 5.2 Multi-Condition Generation Results

For quantitative evaluation, we focus on multiple inputs to image synthesis output since the evaluation metric for this case (FID) does not require specific modality inputs like text. We test with several input combinations including text + image, text + audio, image + audio, text + video, as well as three inputs text + audio + image. We test on the validation set of AudioCaps [24] since all four modalities are present in this dataset. The prompt image input is the middle frame of the video. As shown in

Table 9: CoDi is capable of generating high quality output (image in this case) from various combinations of prompt modalities.

| Inputs | FID ↓ |
|---|---|
| **Single-modality Prompt** | |
| Text | 14.2 |
| Audio | 14.3 |
| **Dual-modality Prompt** | |
| Text + Audio | 14.9 |

Table 10: MSR-VTT text-to-video generation performance.

| Inputs | CLIPSIM ↑ |
|---|---|
| **Single-modality Prompt** | |
| Text | 0.2890 |
| **Dual-modality Prompt** | |
| Text+Audio | 0.2912 |
| Text+Image | 0.2891 |
| Text+Audio+Image | 0.2923 |

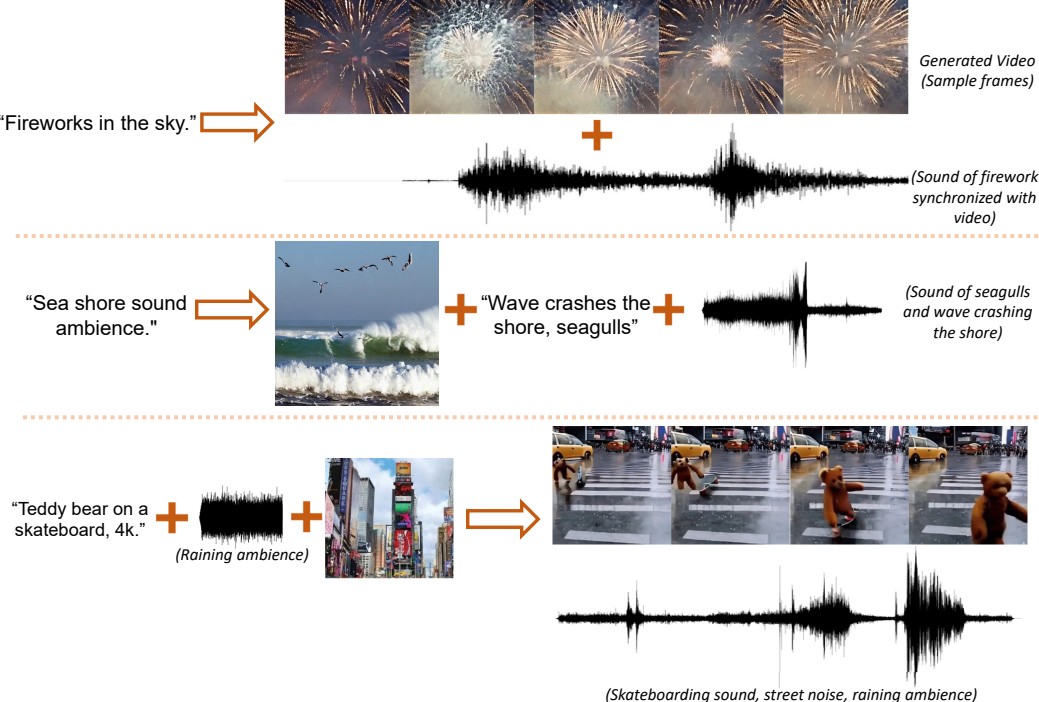

Figure 5: Joint generation of multiple output modalities by CoDi. From top to bottom: text→video+audio, text→image+text+audio, text+audio+image→video+audio.

Table 9, CoDi achieves high image generation quality given assorted groups of input modalities. We also test with several input combinations with video as output including text, text + audio, image + image, as well as text + audio + image. We also test on MSRVTT [24] since all four modalities are present in this dataset. Similarly, the prompt image input is the middle frame of the video. As shown in Table 10, CoDi achieves high video and ground truth text similarity given assorted groups of input modalities. Again our model does not need to train on multi-condition generation like text + audio or text + image. Through bridging alignment and composable multimodal conditioning as proposed in Section 3.2, our model trained on single condition can zero-shot infer on multiple conditions.

### 5.3 Multi-Output Joint Generation Results

For joint multimodal generation, we first demonstrate high-quality multimodal output joint generation demo as shown in Fig. 5. For quantitative evaluation, there is no existing evaluation metric since we are the first model that can simultaneously generate across all 4 modalities. Therefore, we propose the following metric SIM that quantifies the coherence and consistency between the two generated modalities by cosine similarity of embeddings:

$$\text{SIM}(A, B) = \cos\left(C_A(A), C_B(B)\right) \tag{4}$$

Table 11: Similarity scores between generated modalities. The number on the left of "/" represents the similarity score of independent generation, and the right it represents the case of joint generation. Jointly generated outputs consistently show stronger coherence.

| Inputs | SIM-IT | SIM-AT | SIM-VT | SIM-VA |
|---|---|---|---|---|
| **Two Joint Outputs** | | | | |
| Audio → Image+Text | 0.251 / **0.260** | - | - | - |
| Image → Audio+Text | - | 0.244 / **0.256** | - | - |
| Text → Video+Audio | - | - | - | 0.240 / **0.255** |
| Audio → Video+Text | - | - | 0.256 / **0.261** | - |
| **Three Joint Outputs** | | | | |
| Text → Video+Image+Audio | 0.256 / **0.270** | 0.240 / **0.257** | - | 0.240 / **0.257** |
| **Multi-Inputs-Outputs** | | | | |
| Text+Image → Video+Audio | - | - | - | 0.247 / **0.259** |

where $A$, $B$ are the generated modalities, and $C_A$ and $C_B$ are aligned encoders that project $A$ and $B$ to the same space. We use the prompt encoder as described in Section 3.2. This metric aims to compute the cosine similarity of the embedding of two modalities using contrastive learned prompt encoders. Thus, the higher the metric, the more aligned and similar the generated modalities are.

To demonstrate the effectiveness of joint generation, assume the prompt modality is $P$, we compare $\text{SIM}(A, B)$ of $A$ and $B$ generated separately vs. jointly, i.e., $\{P \rightarrow A, P \rightarrow B\}$ vs. $\{P \rightarrow A + B\}$. The benchmark is the validation set of AudioCaps [24]. We test on the following settings, audio →image+text, image →audio+text, and text→video+audio, image →video+audio. audio→ video+text, audio→ text+video+image, text →video+image+audio, where the image prompt is the middle frame of the video clip. As shown in Table 11, joint generation (similarity shown on the right side of "/") consistently outperforms independent generation (on the left side of "/").

# 6 Conclusion

In this paper, we present Composable Diffusion (CoDi), a groundbreaking model in multimodal generation that is capable of processing and simultaneously generating modalities across text, image, video, and audio. Our approach enables the synergistic generation of high-quality and coherent outputs spanning various modalities, from assorted combinations of input modalities. Through extensive experiments, we demonstrate CoDi's remarkable capabilities in flexibly generating single or multiple modalities from a wide range of inputs. Our work marks a significant step towards more engaging and holistic human-computer interactions, establishing a solid foundation for future investigations in generative artificial intelligence.

**Limitations & Broader Impacts.** See Appendix D for the discussion.

# Acknowledgement

We would like to thank Bei Liu for HD-VILA-100M data support. We also thank Shi Dong, Mahmoud Khademi, Junheng Hao, Yuwei Fang, Yichong Xu and Azure Cognitive Services Research team members for their feedback.

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

# A Model Architecture and Configuration

## A.1 Overview

In this section, we provide more details on the model architecture as shown in Table 12, where each modality specific diffuser is based on UNet architecture with different variations detailed in the table. Another notable difference is the video architecture where we add temporal attention and temporal shift as discussed in Section 3.3 and we will discuss its detail in the next section.

Table 12: Hyperparameters for our diffusion models. Note the video and image generation uses the same diffuser.

| Modality | Video (Image) LDM | Audio LDM | Text LDM |
|---|---|---|---|
| **Hyperparameter** | | | |
| Architecture | LDM | LDM | LDM |
| z-shape | $4 \times \#\text{frames} \times 64 \times 64$ | $8 \times 256 \times 16$ | $768 \times 1 \times 1$ |
| Channels | 320 | 320 | 320 |
| Depth | 4 | 2 | 2 |
| Channel multiplier | 1,2,4,4 | 1,2,4,4 | 1,2,4,4 |
| Attention resolutions | 64,32,16 | 64,32,16 | 64,32,16 |
| Head channels | 32 | 32 | 32 |
| Number of heads | 8 | 8 | 8 |
| CA embed dim | 768 | 768 | 768 |
| CA resolutions | 64,32,16 | 64,32,16 | 64,32,16 |
| Autoencoders | AutoKL | AudioLDM | Optimus |
| Weight initialization | Stable Diffusion-1.4 | - | Versatile Diffusion |
| Parameterization | $\epsilon$ | $\epsilon$ | $\epsilon$ |
| Learning rate | $2e-5$ | $5e-6$ | $5e-5$ |
| Total batch size | 256 | 1024 | 1024 |
| **Diffusion Setup** | | | |
| Diffusion steps | 1000 | 1000 | 1000 |
| Noise schedule | Linear | Linear | Linear |
| $\beta_0$ | 0.00085 | 0.00085 | 0.00085 |
| $\beta_T$ | 0.0120 | 0.0120 | 0.0120 |
| **Sampling Parameters** | | | |
| Sampler | DDIM | DDIM | DDIM |
| Steps | 50 | 50 | 50 |
| $\eta$ | 1.0 | 1.0 | 1.0 |
| Guidance scale | 2.0 | 7.5 | 2.0 |

## A.2 Video LDM Architecture

Except for the base image UNet architecture, we also add temporal attention and temporal shift [2] before each residual block. Following VDM [21], the temporal attention is a transformer attention module where we flatten the height and width dimension to batch size dimension and the self-attention is performed on the time dimension. The temporal shift is illustrated in Fig. 6 where we first split channels into $k$ chunks. Then, we shift the channel dimension numbered 0 to $k-1$ by temporal dimension from 0 to $k-1$ times respectively. Eventually, we concatenate the shifted chunks by the hidden dimension. Note that we use $k=3$ in the illustration for simplicity but $k=8$ in our implementation. We then add a convolution layer before the temporal shift module. Finally, we use residual connection [18] and add the output to the input before the convolution layer. The complete video UNet layer is shown in Fig. 7.

# B Model Training

**Prompt Encoders Training.** As discussed in Section 3.2, we use bridging alignment to perform contrastive learning between all prompt encoders. We use Adam [26] optimizer with learning rate 1e-4 and weight decay 1e-4.

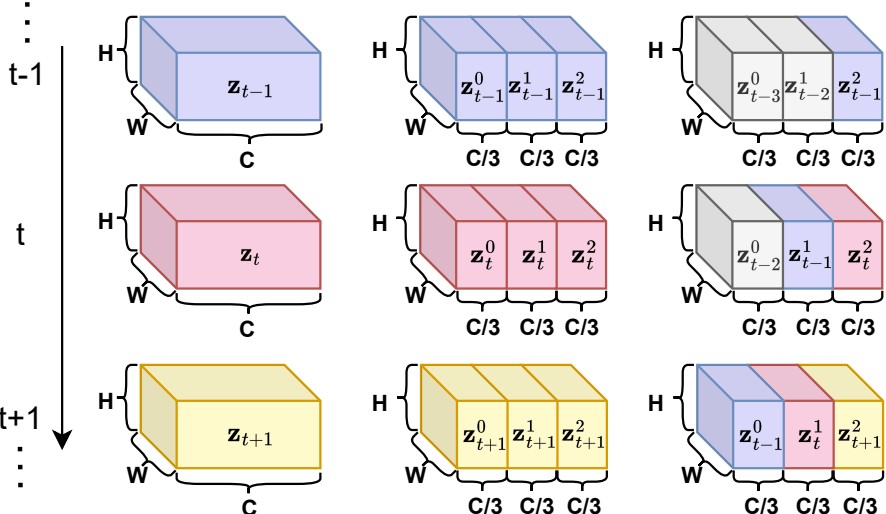

Figure 6: Temporal shift [2] illustration. $C$, $H$, $W$ represent channel, height, width, respectively. The vertical line represents time steps from $t-1$, $t$, and $t+1$. The grey blocks denote "padding tensors".

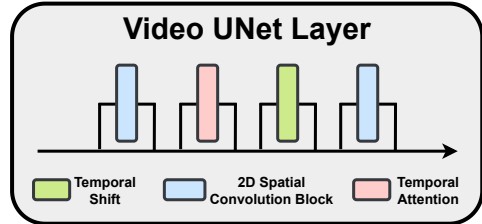

Figure 7: Video UNet layer architecture details including normalization & activation, 2D temporal attention, followed by temporal shift and 1D spatial convolution.

**Diffusion Model Training.** We train diffusion model with training objectives and hyperparameters detailed in Table 1 and Table 12. For video LDM, we adopt a more specific training curriculum. We adopt curriculum learning on frame resolution and frames-per-second (FPS). First, the diffuser is trained on the WebVid dataset of a 256-frame resolution, with the training objective being text-conditioned video generation. The training clips are sampled from 2-second video chunks with 4 FPS. Second, the model is further trained on HDVILLA and ACAV datasets, with a 512-frame resolution and 8 FPS, and the training objective is image-conditioned video generation (the image is a randomly sampled frame of the clip). Each training clip contains 16 frames sampled from a 2-second video chunk with 8 FPS.

**Joint Generation Training.** As discussed in Section 3.2, we train joint generation by aligning environment encoders and optimize cross-attention layers only in the diffusion models. We use Adam optimizer with learning rate 1e-5 and weight decay 1e-4.

## C  Training Datasets

In this section, we introduce more details about the video and audiovisual training datasets.

**Video.** WebVid [4] is a large-scale dataset of web videos with diverse content, spanning over 40 categories such as sports, cooking, and travel. It contains over 1.2 million video clips (all without sound) that are all at least 30 seconds in duration with video descriptions. We perform **text→video** and video-text contrastive learning task with this dataset. HD-Villa-100M [54] is a large-scale video dataset with over 100 million video clips sourced from YouTube. The dataset covers a wide range of video categories and includes high-quality videos with a resolution of at least 720P. Since it lacks

curated video description and we use the middle frame as image input to perform **image→video** generation.

**Audiovisual.** SoundNet originally contains over two million sounds and spans a wide range of categories including music, animal sounds, natural sounds, and environmental sounds. We collected all currently accessible 1M videos.

## D    Limitations & Broader Impacts

While the paper primarily focuses on the technical advancements and potential applications of CoDi, we also consider potential negative social impacts that could arise from the development and deployment of such technology. These impacts can include:

**Deepfakes and Misinformation.** As part of a common issue for generative AI models, the ability of CoDi to generate realistic and synchronized multimodal outputs also raises concerns about the creation and dissemination of deepfakes. Malicious actors could exploit this technology to create highly convincing fake content, such as fabricated videos or audio clips, which can be used for misinformation, fraud, or other harmful purposes.

**Bias and Stereotyping.** If the training data used for CoDi is biased or contains stereotypes, the generated multimodal outputs may also reflect these.

## E    License

We will publicly release our code and checkpoints. We cite licenses from the individual dataset or package we use from the community and provide the following links for references.

**LAION-400M:** Creative Common CC-BY 4.0

**AudioSet:** Creative Common CC-BY 4.0

**AudioCaps:** MIT

**Freesound:** Creative Commons

**BBC Sound Effect:** The BBC's Content Licence

**SoundNet:** MIT

**Webvid10M:** Webvid

**HD-Villa-100M:** Research Use of Data Agreement v1.0

**PyTorch:** BSD-style

**Huggingface Transformers:** Apache

**Torchvision:** BSD 3-Clause

**Torchaudio:** BSD 2-Clause

