# OpenReview forum: "Any-to-Any Generation via Composable Diffusion"
_NeurIPS.cc/2023/Conference — NeurIPS 2023 poster_

### Official Review · Reviewer_hGLR · 2023-06-27

**Soundness:** 4 excellent
**Presentation:** 3 good
**Contribution:** 4 excellent
**Rating:** 7
**Confidence:** 3

**Summary:**

They present Composable Diffusion (CoDi), a novel generative model capable of generating any combination of output modalities, such as language, image, video, or audio, from any combination of input modalities.  Unlike existing generative AI systems, CoDi can generate multiple modalities in parallel and its input is not limited to a subset of modalities like text or image. Despite the absence of training datasets for many combinations of modalities, they also propose to align modalities in both the input and output space.

**Strengths:**

1 One model that takes any combination of modalities as input or output is novel and promising.
2 As the lack of training data, the alignment of different modalities is very difficult. The proposed method for the alignment is very interesting.

**Weaknesses:**

1 The simple weighted interpolation of different representations is not so convincing. Why does this method work?

**Questions:**

see above

**Limitations:**

not addressed

---

> ### Author Rebuttal · Authors · 2023-08-10
>
> Thank you for the review! Please find our response below:
>
> > **1. "Missing discussion of limitation and societal impact."**
>
> The discussion of limitation and societal impact can be found in Section D of the appendix. We attach those paragraphs below:
> Deepfakes and Misinformation: As part of a common issue for generative AI models, the ability of CoDi to generate realistic and synchronized multimodal outputs also raises concerns about the creation and dissemination of deepfakes. Malicious actors could exploit this technology to create highly convincing fake content, such as fabricated videos or audio clips, which can be used for misinformation, fraud, or other harmful purposes.
> Bias and Stereotyping: If the training data used for CoDi is biased or contains stereotypes, the generated multimodal outputs may also reflect these.
>
> > **2. "The simple weighted interpolation of different representations is not so convincing. Why does this method work?"**
>
> The input modalities are encoded by contrastive learned encoders and thus their features representation is aligned and scaled in the same space and distribution. Assuming that the individual embeddings are in a continuous space where distances and directions have meaningful interpretations, the weighted average should also lie within this space. The final representation would maintain these properties, preserving the relationships between concepts.
>
> There are also many advantages and reasons to use simple weighted interpolation:
>
> **Simplicity and explainability:** Weighted averaging is a straightforward operation that's easy to understand, implement. The weighted average approach allows for some level of interpretability, as you can analyze the contribution of each modality by manipulating and observing the effect of the weights.
>
> **Robustness:** If one modality is noisy or missing some information, the other modalities can compensate. The weighted average can be seen as a form of ensemble, potentially providing a more stable and robust representation.
>
> **Flexibility in Emphasizing Modalities:** By adjusting the weights, you can emphasize or de-emphasize certain modalities according to their relevance or reliability for the task at hand.
>
> **Efficiency:** This approach can be computationally efficient, as it doesn't require extensive fine-tuning or additional layers to merge the embeddings.
>
> That said, we agree that it is a meaningful future work direction for creating learned or more complex representations of the contrastive aligned input embeddings to improve the complex interaction of different modalities.

---

> > ### Comment · Reviewer_hGLR · 2023-08-14
> > **Rebuttar Readed**
> >
> > Thanks for your answering and I have read it.

---

### Official Review · Reviewer_mZPw · 2023-07-04

**Soundness:** 3 good
**Presentation:** 3 good
**Contribution:** 3 good
**Rating:** 5
**Confidence:** 5

**Summary:**

This paper presents a method that can generate any combination of output modalities, including language, audio, image, or video, from any combination of input modalities. The idea here is to align four modalities in a shared feature space first, and then learn to generate one or more modalities based on the shared feature space. This design enables many combinations of modalities despite the lack of training datasets. Since the feature space is shared, it also flexible to extend to other modalities.

**Strengths:**

* The idea of any-to-any generation is interesting, and it enables many different tasks in one model.
* The framework is flexible and customizable to many other potential modalities, such as semantic maps, heat map, depth map and so on.
* The performance of the proposed method achieves comparable or better results than previous SOTA methods.


**Weaknesses:**

* The method part is not clear. The relation among image diffusion model, video diffusion model, vision encoder and vision unet is confusing. Since 4 diffusion models are introduced and only 3 types of encoders and unet are shown in Figure 2, It s not clear whether image and video models share the parameters or not.
* The evaluation of Table 3 is not sufficient. Only the text-video faithfulness (CLIPSIM) is evaluated, while the video quality (FVD) is not evaluated.
* The proposed framework enables many different tasks. However, it does not outperform previous SOTA methods in many tasks, such as text-to-video generation, text-to-image generation, image captioning and video captioning.


**Questions:**

* From Table 8, using both text and audio as input achieves higher FID compared to using each single modality as input. Could you explain why model achieves worse performance with more information as input?
* From table 2 and table 3, CoDi does not outperform previous SOTA results. Do you think a model that can do all tasks need to sacrifice its performance on each specific task?
* During training, the text encoder weights are frozen after training with images, would it result to a suboptimal problem when training with other modalities?
* In Sec 3.3, image diffusion model and video diffusion model are introduced separately. However, in Figure 2, only vision UNet and Vision Encoder are shown. Does it mean image diffusion model share parameters with video diffusion model during training?
* In table 4, why CoDi can outperform other diffusion-based method in image captioning?

**Limitations:**

The authors adequately address the limitations and potential negative sosietal impact.

---

> ### Author Rebuttal · Authors · 2023-08-10
>
> Thank you for the insightful review. We are committed to addressing the concerns raised to enhance the quality of this paper.
>
> > **1. "The method part is not clear. The relation among image diffusion model, video diffusion model, vision encoder and vision unet is confusing. Since 4 diffusion models are introduced and only 3 types of encoders and unet are shown in Figure 2, It’s not clear whether image and video models share the parameters or not."**
>
> In line number 148-149, we stated that we construct the video diffuser by extending the image diffuser with temporal modules. This implies that the image and video generation share the same model. Concretely, both the encoder and UNet use the same architecture for image and video. For vision encoder, we add temporal layers on top of the image encoder for video encoding. Similarly for vision UNet, we also add temporal layers on top of the image UNet for video generation. In the Appendix A, we have also discussed the detailed video architecture. We will make this clearer in the paper.
>
> > **2. "The evaluation of Table 3 is not sufficient. Only the text-video faithfulness (CLIPSIM) is evaluated, while the video quality (FVD) is not evaluated."**
>
> In the table below, on video generation on UCF-101, CoDi performs competitively with state-of-the-art which matches similar observation in CLIPSIM metric conducted on MSRVTT.
>
> | Method                             | Zero-Shot | IS ($\uparrow$ )  | FVD ($\uparrow$ )  |
> |------------------------------------|-----------|---------|----------|
> | CogVideo (Chinese) | Yes       | 23.55   | 751.34   |
> | CogVideo (English) | Yes       | 25.27   | 701.59   |
> | Make-A-Video          | Yes       | 33.00   | 367.23   |
> | Video LDM         | Yes       | 33.45   | 550.61   |
> | **CoDi (Ours)**                    | Yes       | 32.88   | 596.34   |
>
>
> > **3. "The proposed framework enables many different tasks. However, it does not outperform previous SOTA methods in many tasks, such as text-to-video generation, text-to-image generation, image captioning and video captioning."**
>
> Please see our response in General Response 2 for full discussion. We’d like to reiterate that the main focus of CoDi is not to beat previous text-to-X SOTA methods, because many of them are proprietary and use private training data. In fact, our audio and text diffusion model achieves SOTA performance, and the image diffusion model (DM) uses the best open-sourced one (Stable Diffusion 1.5) at the time of this project. We acknowledge that CoDi’s video DM has a gap between previously-reported SOTAs (such as imagegen-vide, make-a-video), however, those DMs are closed-sourced and trained on private data (such as company-internal videos). In contrast, CoDi’s video DM is trained on public data and open-sourced.
>
> The main focus of CoDi is enabling different pretrained DMs to communicate and interact with each other, such that we can achieve any-to-any generation even without the existence of paired training data.
>
> Moreover, CoDi is designed with composability and modularity at its core. This means that it can readily incorporate various diffusion models into the framework without requiring a significant amount of retraining. This advantageous quality ensures that CoDi remains agile and adaptable, allowing it to seamlessly leverage many state-of-the-art (SOTA) diffusion models as they become available.
>
> > **4. "From Table 8, using both text and audio as input achieves higher FID compared to using each single modality as input. Could you explain why the model achieves worse performance with more information as input?"**
>
> Please see our response in General Response 2 for full discussion. In general, the primary goal of the model under discussion was to demonstrate the ability to handle multiple modalities without significant performance degradation. The slight increase in FID scores from 14.2 to 14.9 is not seen as a meaningful degradation, especially given the statistical significance level (p=0.086), and this trade-off is justified by the model's increased flexibility and broader application potential. Additionally, CLIPSIM metrics affirm the model's faithfulness in video generation to the input text, even when adding audio modalities. Table 10 further illustrates CoDi's effectiveness in integrating different input modalities, showing clear improvement for video and audio joint generation when adding the image modality.
>
> > **5. "During training, the text encoder weights are frozen after training with images, would it result to a suboptimal problem when training with other modalities?"**
>
> The proposed method bridging alignment leverages the property that the text encoder is frozen. There are several advantages or reasons:
>
> **Efficiency:** The training is much more efficient (without finetuning the text encoder and jointly training all combinations of contrastive learning). Finetuning the text encoder will result in double the cost for all encoder training since text encoder is the bridging encoder that participates in all training.
>
> **Data scale:** The text and image models are trained with a very large scale dataset with 400M which has very strong generalization potential. Such joint image and text embedding learned will be sufficient to be extended to other modalities. On the other hand, the text-audio and text-video dataset has a much smaller scale with only a few million samples.
>
> **Learnable diffusion:** Regardless of the contrastive learned encoders’ performance, the diffusion model cross attention is trainable and will adapt its distribution to the desired or target tasks.
>
> > **6. "In table 4, why CoDi can outperform other diffusion-based method in image captioning?"**
>
> We use LAION-400M as the training data whereas previous works like SCD-Net shown in the table 4 uses a much smaller dataset, MSCOCO, which has below 1M examples. Training on significantly more data with similar architecture will result in better performance.

---

> > ### Comment · Reviewer_mZPw · 2023-08-10
> >
> > For question 2, it's necessary to provide FVD results on MSRVTT to compare video qualities with other methods. Competitive CLIPSIM does not guarantee competitive FVD. Competitive FVD on UCF101 also does not guarantee competitive FVD on MSRVTT.

---

> > > ### Author Response · Authors · 2023-08-11
> > >
> > > Thank you for your review.
> > >
> > > We acknowledge the importance of comparing our method with previous work using the FVD metric on MSR-VTT. However, almost no previous work tested FVD on MSRVTT. Plus most of these works are not open-source and unable to test. CogVideo stands out as an exception, being the only open-source video generation model (that is already in Table 3) available, and as such, we have included its FVD score in our table.
> > >
> > > Below, please find the extended Table 3 showcasing MSR-VTT text-to-video generation performance, now including FVD scores.
> > >
> > > ### Table 3: MSR-VTT text-to-video generation performance extended to FVD.
> > >
> > > | Method          | Zero-Shot | CLIPSIM $\uparrow$ | FVD $\downarrow$ |
> > > |-----------------|:---------:|:---------:|:---------:|
> > > | GODIVA    | No        | 0.2402    | _ |
> > > | NÜWA       | No        | 0.2439    | - |
> > > | CogVideo   | Yes       | 0.2631    | 801.98 |
> > > | Make-A-Video | Yes      | 0.3049    | - |
> > > | Video LDM   | Yes       | 0.2929    | - |
> > > | CoDi (Ours)     | Yes       | 0.2890    | 612.02 |

---

### Official Review · Reviewer_dG2H · 2023-07-06

**Soundness:** 3 good
**Presentation:** 3 good
**Contribution:** 3 good
**Rating:** 6
**Confidence:** 4

**Summary:**

The paper introduces Composable Diffusion (CoDi), an innovative generative model capable of producing any combination of output modalities, such as language, image, video, or audio, from any combination of input modalities. Unlike existing generative AI systems, CoDi can generate multiple modalities simultaneously and is not limited to a subset of modalities like text or images. To address the challenge of lacking training datasets for many modalities combinations, the authors propose a modality alignment approach in both the input and output space. This enables CoDi to condition freely on any input combination and generate any group of modalities, even if they are not present in the training data. CoDi employs a unique composable generation strategy that establishes a shared multimodal space through alignment in the diffusion process. This allows for the synchronized generation of intertwined modalities, such as temporally aligned video and audio. Offering high customization and flexibility, CoDi achieves impressive quality in joint-modality generation and either outperforms or matches the state-of-the-art unimodal models for single-modality synthesis.

**Strengths:**

1. The paper is addressing an important problem of mapping modalities from any domain to any domain without fully paired data.
2. The proposed method is novel and reasonable. It is good to see that each different component can be trained separately.
3. The proposed bridging alignment is interesting.

**Weaknesses:**

The proposed method shares some similarities with previous works. Nevertheless, this paper still contributes to the community in my opinion. It could be better to have a more specific discussions on the difference with the related work.

**Questions:**

Please refer to weakness.

**Limitations:**

Yes.

---

> ### Author Rebuttal · Authors · 2023-08-10
>
> Thanks, we agree that specific comparisons can provide more insights and clarify the unique contributions of CoDi. Here's how we position our work with different areas and community in extension to the the discussion in the related work section:
>
> >**1. "Comparison with previous diffusion models."**
>
> We first provide a context to the current diffusion models community, in addition to the diffusion background introduced in 66-81.
>
> **Scope:** Most diffusion models focus on specific generative tasks, such as text-to-X or single-to-single generation. Some examples are text-to-video, text-to-image, image/audio/video captioning, etc. In contrast, our work ambitiously aims to build a versatile framework for any-to-any generation within the realms of image, text, audio, and video.
>
> **Multi-modal Integration:** Previous works [1,2,3,4] that involve multi-modal generation often lack a unified encoding space [1] or a method to maintain high correspondence between output modalities effectively [2,3]. CoDi innovatively addresses these limitations by proposing bridging alignment that facilitates the integration of various input modalities and ensures high correspondence in the generated outputs.
>
> [1] Liu, Haotian, et al. "Visual instruction tuning." arXiv preprint arXiv:2304.08485 (2023).
>
> [2] Xu, Xingqian, et al. "Versatile diffusion: Text, images and variations all in one diffusion model." arXiv preprint arXiv:2211.08332 (2022).
>
> [3] Masahiro Suzuki and Yutaka Matsuo. A survey of multimodal deep generative models. Advanced Robotics, 36(5- 6):261–278, 2022.
>
> [4] Wu, Mike, and Noah Goodman. "Multimodal generative models for compositional representation learning." arXiv preprint arXiv:1912.05075 (2019).
>
> >**2. "Comparison with previous general multimodal frameworks."**
>
> We also provide a broader context to other general multimodal frameworks like BLIP [5], Flamingo [6], Llava [1], etc, in addition to the multimodal background introduced in 82-89.
>
> **Focus on Generation vs. Reasoning:** While these frameworks emphasize multi-modal reasoning tasks such as question answering or dialogue, CoDi's primary goal is to enable flexible generation from any combination of input modalities. This distinct focus sets our work apart and addresses a different set of challenges and opportunities.
>
> **Bridging Alignment Contribution:** A key innovation in CoDi is the design of bridging alignment, which reduces the quadratic training/data cost to linear, enabling efficient handling of unpaired data. This technical advancement further distinguishes our work and has broad implications for efficiency and scalability.
>
> We recognize the value in a more comprehensive discussion with related works and will indeed include this in the revised manuscript. This discussion will provide readers with a deeper understanding of CoDi's place within the existing landscape and highlight our novel contributions more prominently.
>
> [5] Li, Junnan, et al. "Blip: Bootstrapping language-image pre-training for unified vision-language understanding and generation." International Conference on Machine Learning. PMLR, 2022.
>
> [6] Alayrac, Jean-Baptiste, et al. "Flamingo: a visual language model for few-shot learning." Advances in Neural Information Processing Systems 35 (2022): 23716-23736.

---

> > ### Comment · Reviewer_dG2H · 2023-08-19
> >
> > Thanks for the response. I have no other questions.

---

### Official Review · Reviewer_7M7p · 2023-07-07

**Soundness:** 2 fair
**Presentation:** 2 fair
**Contribution:** 3 good
**Rating:** 6
**Confidence:** 4

**Summary:**

The paper presents a new generative model called Composable Diffusion (CoDi). This model is capable of generating any combination of output modalities from any combination of input modalities, including language, image, video, or audio. Unlike other models that are limited to a subset of modalities like text or image, CoDi can generate multiple modalities in parallel.

The authors have designed CoDi to align modalities in both the input and output space. This allows the model to condition on any input combination and generate any group of modalities, even if they are not present in the training data.

A key feature of CoDi is its novel composable generation strategy. This involves building a shared multimodal space by bridging alignment in the diffusion process. This feature enables the synchronized generation of intertwined modalities, such as temporally aligned video and audio.

The paper reports that CoDi achieves strong joint-modality generation quality. It either outperforms or is on par with the unimodal state-of-the-art for single-modality synthesis.

**Strengths:**

1. Originality: The paper introduces Composable Diffusion (CoDi), a new model in multimodal generation. This model is designed to process and generate modalities across text, image, video, and audio simultaneously. This is a novel contribution as it enables the generation of various output modalities from different combinations of input modalities.

2. Quality: The authors have conducted extensive experiments to demonstrate the capabilities of CoDi. The results show that CoDi can generate single or multiple modalities from a wide range of inputs. The model's performance is competitive with state-of-the-art models in tasks such as image and video generation, video captioning, and image synthesis from multiple input modalities.

3. Clarity: The paper is well-structured and provides clear explanations of the model's architecture and its generation strategy. The use of figures and tables helps to understand the model's capabilities and performance.

4. Significance: This work represents a step towards more comprehensive human-computer interactions by enabling the generation of multiple modalities in parallel. CoDi has potential applications in various areas, from content creation to human-computer interaction. The authors also provide a basis for future research in generative artificial intelligence.

In summary, the paper presents a significant and original contribution to the field of multimodal generation, demonstrating high-quality research and clear presentation.

**Weaknesses:**

The paper presents a novel approach to multimodal generation, but there are several areas where it could be improved:

1. Evaluation Metrics: The evaluation of the model's performance is primarily based on quantitative metrics such as Frechet Inception Distance (FID) and CLIPSIM. These metrics, while useful, may not fully capture the perceptual quality or coherence of the generated outputs. Incorporating user studies or other qualitative evaluations could provide a more comprehensive understanding of the model's performance.

2. Quality of Generated Results: The quality of the generated results could be improved. The generated videos are relatively short, the quality of the images is perceptually low, and the generated text is often short and discontinuous. These factors could limit the practical utility of the generated outputs.

3. Preservation of Input Modality: The model primarily focuses on understanding between modalities, but it does not always preserve the faithfulness of the input modality. For instance, the output video and images do not consistently preserve the identity of the input image. This could limit the model's ability to generate accurate and coherent outputs across different modalities.

4. Cross-Modality Benefits: The paper does not convincingly demonstrate that the generation results benefit from cross-modality conditions. For example, Table 8 shows that the quality of image generation can even degrade when using conditions from two modalities. Similarly, Table 9 shows only marginal improvements in video quality when using multiple modalities. The authors should establish a benchmark that clearly demonstrates the benefits of using multiple modalities for generation. Without such evidence, the necessity of the proposed architecture could be questioned.

5. Omission of Baselines: In Table 2, the authors omit the StableDiffusion v1.5 baseline, which is the image Latent Diffusion Model (LDM) they used. Including this baseline could provide a more comprehensive comparison of the model's performance.

**Questions:**

1. Evaluation Metrics: Could you provide more details on why you chose FID and CLIPSIM as the primary evaluation metrics? Have you considered incorporating user studies or other qualitative evaluations to assess the perceptual quality and coherence of the generated outputs?

2. Quality of Generated Results: Could you elaborate on the factors that might be contributing to the short and discontinuous text, short video length, and perceptually low-quality images? Are there potential improvements or modifications to the model that could address these issues?

3. Preservation of Input Modality: How does the model ensure the preservation of the identity or characteristics of the input modality in the generated outputs? Are there specific mechanisms in place to ensure this, or is it an area for future work?

4. Cross-Modality Benefits: Could you provide more evidence or a clearer explanation of how the generation results benefit from cross-modality conditions? The results in Tables 8 and 9 suggest that the benefits might be marginal or even negative in some cases. Could you clarify this?

5. Omission of Baselines: Why was the StableDiffusion v1.5 baseline omitted from the comparisons in Table 2? Including this baseline could provide a more comprehensive view of the model's performance relative to existing methods.

**Limitations:**

 The authors have adequately addressed the limitations and potential negative societal impact of their work.

---

> ### Author Rebuttal · Authors · 2023-08-10
>
> Thank you for the review.
>
> > **1. "Evaluation Metrics: Incorporating user studies or other qualitative evaluations could provide a more comprehensive understanding of the model's performance."**
>
> We perform a small scale user study due to time constraints. We will perform more comprehensive studies in the next version of the paper.
>
> **Text to Audio**
>
> We first conduct text to audio user study by comparing CoDi to previous text-to-audio generation SOTA AudioLDM on 30 text-to-audio generation examples.
>
> For the user 0: CoDi was favored in 23 instances. AudioLDM was favored in 7 instances.
>
> For the user 1: CoDi was favored in 18 instances. AudioLDM was favored in 12 instances.
>
> For the user 1: CoDi was favored in 21 instances. AudioLDM was favored in 9 instances.
>
> Overall, the three users clearly showed a preference for the CoDi method for text-to-audio generation.
>
> **Audio to Image + Text**
>
> We next conduct audio to image+text user study by comparing CoDi with joint generation to CoDi without joint generation capacity.
>
> For the user 0: CoDi without joint generation was favored in 10 instances. CoDi with joint generation was favored in 20 instances.
>
> For the user 1: CoDi without joint generation was favored in 10 instances. CoDi with joint generation was favored in 20 instances.
>
> For the user 2: CoDi without joint generation was favored in 8 instances. CoDi with joint generation was favored in 22 instances.
>
> Overall, the three users clearly showed a preference for the CoDi method with joint generation.
>
> > **2. "Quality of Generated Results: The quality of the generated results could be improved."**
>
> Please see General Response 1 for full discussion. In general, CoDi focuses on multi-modal generation and enabling different pretrained DMs to communicate and interact with each other. We only maximally maintain competitive single generation performance without sacrificing individual task performance. Previous SOTA diffusion models are often proprietary and use private data while our model is fully open-source. Moreover, the individual diffusion models are already very competitive and some achieve SOTA performance as shown in tables 2-7.
>
> > **3. "Preservation of Input Modality: The model primarily focuses on understanding between modalities, but it does not always preserve the faithfulness of the input modality."**
>
> Thank you for the observation regarding the preservation of input modality. To this end, we have experimented with finetuning CoDi for image animation, i.e., input an image and generate a video that animates it. See **examples in the uploaded pdf**. The **generated videos** are also shared with the AC. Concretely, on top of the original CoDi video diffuser that takes in encoder embeddings, we concatenate the image to the diffuser inputs to improve faithfulness of image animation. This shows that CoDi can be easily modified or finetuned to perform other downstream tasks with higher faithfulness.
>
> In general, faithfulness, or modality preservation, is an ongoing challenging topic in the diffusion model and AIGC community. Still, it is a meaningful future work to modify the architecture and training on specific tasks where faithfulness to input images for example can be preserved.
>
> The design choices of CoDi does not focus on preserving the faithfulness of the input modality because of the following reasons:
>
> (1) Focus on Modality Conversion: CoDi's primary goal is to explore modality conversion, enabling seamless transformation between various modalities like text, images, audio, and video. It's designed to handle flexible and innovative tasks, rather than strictly preserving the input modality. This approach can lead to creative and adaptive generation capabilities.
>
> (2) Use Cases: The preservation of input modality may be context-dependent. In some applications or use cases, a higher degree of preservation may be desirable, while in others, more flexible and transformative generation may be preferred. Our model is designed to be versatile, and potential adjustments to emphasize input preservation could be explored depending on specific requirements or user needs.
>
> > **4. "Cross-Modality Benefits: The paper does not convincingly demonstrate that the generation results benefit from cross-modality conditions."**
>
> Please see General Response 2, where we discuss the difference in FID scores and CLIPSIM between single and multi-modal approaches in a model. The main goal was to show that the model can handle multiple modalities without a significant drop in performance, rather than achieving a lower FID score. Though there is a slight increase in FID scores, it's not statistically significant (p = 0.086) and is outweighed by the model's increased flexibility and potential applications. CLIPSIM, a metric for video generation fidelity, shows that adding audio modalities as input conditions maintains similarity between video output and text input. The data in Table 10 further supports the effectiveness of integrating different input modalities using CoDi for joint video and audio generation (row Text -> Video+Audio (0.240 / 0.255) and row Text + Image -> Video+Audio (0.247 / 0.259),).
>
> > **5. "Omission of StableDiffusion v1.5 baseline"**
>
> In the table below, we compare our work to StableDiffusion v1.5 and still performs comparably. And note that CoDi is composable and modular. Therefore, it will be very efficient to add other diffusion models to the framework without a significant amount of training. Such advantage allows CoDi to take advantage of many SOTA diffusion models.
>
> | **Method**                    | **FID $\downarrow$** |
> |-------------------------------|-----------|
> | CogView | 27.10     |
> | GLIDE   | 12.24     |
> | Make-a-Scene | 11.84  |
> | LDM    | 12.63     |
> | Stable Diffusion-1.4          | 11.21     |
> | Stable Diffusion-1.5          | 11.12     |
> | Versatile Diffusion | 11.10 |
> | **CoDi (Ours)**               | 11.26     |

---

### Author Rebuttal · Authors · 2023-08-10

We are glad all reviewers appreciated our work and found it well-motivated (7M7p, dG2H, mZPw, hGLR), well-written (7M7p, dG2H, mZPw), and original in introducing Composable Diffusion as a novel model (7M7p, dG2H, mZPw, hGLR). The recognition of CoDi's capability to generate any combination of output modalities and the novel method of alignment (7M7p, dG2H, hGLR), the extensive experiments demonstrating CoDi's competitive performance with state-of-the-art models (7M7p, mZPw), the clear structure and explanation of the model's architecture (7M7p), and the idea of any-to-any generation that provides flexibility and customization (mZPw) were encouraging. Furthermore, the acknowledgment of CoDi's potential applications in various areas and its significance as a step towards more comprehensive human-computer interactions (7M7p) validates the impact of our work. We appreciate the thorough assessment and constructive feedback from all reviewers.

>1. **General Response 1 (the performance of each individual diffusion model)**:

We thank the reviewers for their feedback on the quality of the generated results.

It is essential to clarify the primary objectives of CoDi. Our key contribution lies in enabling different pretrained diffusion models to communicate and interact with each other. The goal is to establish a framework that facilitates this interaction, rather than fine-tuning individual aspects of unimodal diffusion models.

In building CoDi, we maximally leverage all the open source models including StableDiffusion, CLIP, etc, for better reproduction and open access. We train our own audio, text, video diffusion models. All the training data we used is open sourced. We devote our resources to explore joint generation and generation from multiple modalities input and only aim to maintain a reasonably competitive single generation performance.

**Videos**: It is worth noting that at the time of doing this project, previously reported video diffusion models SOTA are closed-sourced (imagegen-video, gen-2, etc), and trained on internal video data. While the generated videos are acknowledged to be relatively short, our video diffusion model demonstrates competitive performance, aligning with SOTA video diffusion models as shown in Table 3. Generating short videos as demos are common approaches of the video community [1,2,3]. Most video models can be extended to longer length by further finetuning and modifications of the architecture and autoregressively generating the video as also shown in [1,2,3]. Still, generating long and coherent video is an ongoing and challenging topic in the video generation community.

[1] Hong, Wenyi, et al. "Cogvideo: Large-scale pretraining for text-to-video generation via transformers." arXiv preprint arXiv:2205.15868 (2022).

[2] Khachatryan, Levon, et al. "Text2video-zero: Text-to-image diffusion models are zero-shot video generators." arXiv preprint arXiv:2303.13439 (2023).

[3] Blattmann, Andreas, et al. "Align your latents: High-resolution video synthesis with latent diffusion models." Proceedings of the IEEE/CVF Conference on Computer Vision and Pattern Recognition. 2023.1.5

**Images**: Regarding the perceived low quality of the images, our model is initialized from Stable Diffusion 1.5, the best open-source image diffusion model available at the time of submission. This foundation provides a robust starting point, and we welcome future work to enhance visual quality within the constraints of our novel cross-modal framework.

**Text**: The generated text primarily serves as a caption, encapsulating the core information in the audio/video/image. This brevity is intentional and in line with common practices in captioning. We have showcased SOTA performance on diffusion-based image captioning, SOTA performance on audio captioning, and competitive performance on video captioning in Tables 4, 5, and 6.

>2. **General Response 2 (multiple inputs performance)**:

We appreciate the observation regarding the difference in FID scores and CLIPSIM between the single modality and multi-modal approaches. However, our primary goal was to demonstrate that our model can successfully handle multiple modalities without significant performance degradation, rather than to achieve a lower FID score per se. The observed difference in FID scores from 14.2 to 14.9, while present, does not constitute a meaningful degradation (statistical significance with only (p=0.086, (0.05<p<0.1)), especially considering the expanded capabilities of handling multiple input types. This trade-off in performance is outweighed by the increased flexibility and potential applications that our model offers.

CLIPSIM is an evaluation metric that evaluates how faithful the video generation is to the input text. We can see in table 9 that by adding audio modalities as input conditions, the similarity between the video output and text input does not decrease, showing the effectiveness of how CoDi can integrate different input modalities while being faithful to each.

In Table 10, row Text -> Video+Audio (0.240 / 0.255) and Text + Image -> Video+Audio (0.247 / 0.259), we can see a clear improvement for video and audio joint generation by adding the image modality. This further supports the effectiveness of CoDi on integrating different input modalities.

---

### Decision · Program_Chairs · 2023-09-21

**Decision:**

Accept (poster)

**Comment:**

This paper was reviewed by four experts in the field. Based on the reviewers' feedback, the decision is to recommend the paper for acceptance to NeurIPS 2023. The reviewers did raise some valuable concerns that should be addressed in the final camera-ready version of the paper. The authors are encouraged to make the necessary changes to the best of their ability. We congratulate the authors on the acceptance of their paper!